# Crabp1 Modulates HPA Axis Homeostasis and Anxiety-like Behaviors by Altering FKBP5 Expression

**DOI:** 10.3390/ijms222212240

**Published:** 2021-11-12

**Authors:** Yu-Lung Lin, Chin-Wen Wei, Thomas A. Lerdall, Jennifer Nhieu, Li-Na Wei

**Affiliations:** Department of Pharmacology, University of Minnesota Medical School, Minneapolis, MN 55455, USA; yllin@umn.edu (Y.-L.L.); wei00170@umn.edu (C.-W.W.); lerda013@umn.edu (T.A.L.); nhieu001@umn.edu (J.N.)

**Keywords:** Crabp1, anxiety, HPA axis, retinoic acid, stress, FKBP5, feedback inhibition

## Abstract

Retinoic acid (RA), the principal active metabolite of vitamin A, is known to be involved in stress-related disorders. However, its mechanism of action in this regard remains unclear. This study reports that, in mice, endogenous cellular RA binding protein 1 (Crabp1) is highly expressed in the hypothalamus and pituitary glands. Crabp1 knockout (CKO) mice exhibit reduced anxiety-like behaviors accompanied by a lowered stress induced-corticosterone level. Furthermore, CRH/DEX tests show an increased sensitivity (hypersensitivity) of their feedback inhibition in the hypothalamic–pituitary–adrenal (HPA) axis. Gene expression studies show reduced FKBP5 expression in CKO mice; this would decrease the suppression of glucocorticoid receptor (GR) signaling thereby enhancing their feedback inhibition, consistent with their dampened corticosterone level and anxiety-like behaviors upon stress induction. In AtT20, a pituitary gland adenoma cell line elevating or reducing Crabp1 level correspondingly increases or decreases FKBP5 expression, and its endogenous Crabp1 level is elevated by GR agonist dexamethasone or RA treatment. This study shows, for the first time, that Crabp1 regulates feedback inhibition of the the HPA axis by modulating FKBP5 expression. Furthermore, RA and stress can increase Crabp1 level, which would up-regulate FKBP5 thereby de-sensitizing feedback inhibition of HPA axis (by decreasing GR signaling) and increasing the risk of stress-related disorders.

## 1. Introduction

Retinoic acid (RA) is the principal active metabolite of vitamin A and plays important roles in multiple biological systems. In the central nervous system, RA is best known to regulate neurogenesis and synaptic plasticity during developmental stages. Clinical and pre-clinical studies have shown that retinoids may also contribute to stress-related disorders, such as depression and anxiety. For instance, hypervitaminosis A is associated with psychiatric disorders such as depression, aggression, and psychosis [1,2,3]. The compound 13-cis RA (isotretinoin; ITT) is an FDA-approved drug for severe acne, but has serious side effects including depression, suicidal ideation and psychosis [3]. Furthermore, chronic treatment with RA or ITT induces anxiety- and depression-like behaviors and impairs homeostasis of the hypothalamic–pituitary–adrenal (HPA) axis in preclinical studies [4]. All the evidence suggests that excess RA can contribute to stress-related disorders, likely through disrupting HPA axis regulation. However, the underlying mechanism is poorly understood. 

The HPA axis is critical for maintaining endocrine system homeostasis, as well as a host of important physiological processes/activities such as immunity, stress responses, and a healthy psychological state, etc. Most notably, under stress, the hypothalamus gland secretes both corticotropin releasing hormone (CRH) and arginine vasopressin (AVP) from the paraventricular nucleus to stimulate the production of adrenocorticotropic hormone (ACTH) in the anterior pituitary gland. ACTH is then released into the circulation to stimulate adrenal cortex to secrete glucocorticoids (corticosterone in mice; cortisol in humans) which regulate a vast array of physiological processes [5,6]. The levels of circulating glucocorticoids are tightly controlled by feedback inhibition via proper activation of glucocorticoid receptor (GR) signaling in the hypothalamus and pituitary glands. This is due to activated GR signaling suppressing the expression of CRH, ACTH, and subsequently glucocorticoid secretion, thereby maintaining systemic homeostasis of this important hormone. Deregulation in the HPA axis can cause numerous health issues, including anxiety and depression [7,8], and stress is best known to disturb the homeostatic control of the HPA axis. A key regulatory event in stress-responses involves the suppression of GR signaling via FK506-binding protein 51 (FKBP5). FKBP5 is a hsp90-associated co-chaperone and provides a potential therapeutic target for stress-related disorders (depression, post-traumatic stress disorder), metabolic disorders (obesity and diabetes) and chronic pain [9,10].

RA is known to bind to nuclear RA receptors (RARs) to regulate gene expression, a well-established canonical activity of RA. Recently, studies have begun to reveal that, Cellular RA-Binding Protein 1 (CRABP1), a highly conserved cytosolic protein with a high RA-binding affinity, can modulate multiple signaling pathways in the cytoplasm, referred to as non-canonical activities of RA, because these activities are independent of RARs [11]. CRABP1-mediated actions of all-*trans* retinoic acid (atRA) typically provide rapid modulation of various cytosolic signaling pathways in different cellular contexts [11]; these non-canonical activities include modulating the activation of ERK and CaMKII. Clinically, disturbing the expression of Crabp1 has been associated with human diseases including cancers and neurological diseases [12,13,14,15,16,17,18,19]. Classical molecular studies have revealed that *Crabp1* gene expression is tightly regulated, particularly sensitive to hormonal fluctuation and epigenetic silencing, and its promoter/enhancer region contains multiple CpG islands and regulatory elements [15]. The tight regulation of the *Crabp1* gene supports the notion that Crabp1 level is important in normal physiological processes.

To this end, it is interesting that acute high-dose synthetic glucocorticoid administration in a dental procedure can significantly reduce the extent of methylation on several CpG islands in the *Crabp1* gene promoter, suggesting that GR or acute stress can upregulate Crabp1 expression [15]. However, it has never been experimentally tested or demonstrated whether and how Crabp1 plays a role in stress-related disorders.

In this study, we set up experiments to investigate the role of Crabp1 in the acute stress response and HPA axis regulation. In studying a Crabp1 knockout (CKO) mouse model, we first found that CKO mice exhibited reduced anxiety-like behaviors and a significantly lowered stress-induced corticosterone level following restraint stress. This was attributable, at least partially, to enhanced feedback inhibition in the HPA axis of CKO mice. To determine the mechanism, we used a pituitary gland cell line, AtT20, which expressed Crabp1 endogenously. In this physiologically relevant in vitro model, we found that lowering Crabp1 level resulted in down-regulated FKBP5 (a GR signaling suppressor) expression. This would enhance GR signaling and, consequentially, a stronger feedback inhibition in animals. Importantly, RA and GR agonists such as Dexamethasone could both increase Crabp1 gene expression in this cell culture model, providing a potential molecular link of Crabp1 to the pathogenesis of certain stress-related disorders.

## 2. Results

### 2.1. Crabp1 Knockout (CKO) Mice Were Less Anxious

In order to understand the role of Crabp1 in the stress response, we employed CKO mice in this study. We first performed open field and elevated plus maze tests to assess anxiety-like behaviors of WT and CKO mice (Figure 1). In the open field test (Figure 1A–C), CKO mice spent more time exploring the central areas (center) of the open field (Figure 1B; T(25) = 2.892, *p* < 0.01), with no difference in the total distance traveled (Figure 1C; T(25) = 1.686, *p* = 0.104). In the elevated plus maze test (Figure 1D–F), CKO mice spent a longer time (increased duration) (Figure 1B; T(25) = 3.665, *p* < 0.01) and with significantly more entries (Figure 1B; T(25) = 2.496, *p* < 0.05) onto the open arms. These results indicated that, without Crabp1, animals were less anxious as reflected in their reduced anxiety-like behaviors.

### 2.2. Deleting Crabp1 Enhanced Feedback Inhibition of HPA Axis

Given the apparent behavioral changes of CKO mice, we then assessed the levels of the major stress hormone, corticosterone, under the control (baseline) condition or restraint stress (Figure 2A,B). As compared to WT mice, CKO corticosterone basal level was reduced by approximately 50%, but this difference did not reach significance (Figure 2A; T(12) = 1.484, *p* = 0.164). However, after 15 min of restraint stress, CKO mice had a significantly lower corticosterone level as compared to WT mice (Figure 2B; T(11) = 3.184, *p* < 0.01). This result is consistent with the behavior data shown in Figure 2A,B, that CKO mice displayed a weaker stress response. We further examined whether regulation of the HPA axis in CKO mice was altered, using the DEX/CRH test which monitored the efficiency of animals’ feedback inhibition in the HPA axis (Figure 2C,D). Dexamethasone (DEX) is a GR agonist that activates GR signaling in the hypothalamus and pituitary glands to suppress the production of CRH and ACTH from these glands. The data showed that, DEX alone could suppress serum corticosterone level in both WT and CKO mice, indicating that feedback inhibition remained functional in CKO mice (Figure 2C; T(15) = 1.222, *p* = 0.241). Interestingly, with DEX + CRH treatment (CRH would induce the secretion of corticosterone), there was a significant (35%) reduction in the corticosterone level of CKO mice as compared to WT mice (Figure 2D; T(15) = 3.648, *p* < 0.01). These results suggested that deleting Crabp1 enhanced feedback inhibition in the HPA axis, which could lower circulating stress hormone levels.

### 2.3. Deleting Crabp1 Reduced the Expression of FKBP5 in Hypothalamus and Pituitary Glands

In order to determine the potential target of Crabp1, in the HPA axis, we first monitored Crabp1 protein level in the relevant glands of the HPA axis (Figure 3A). It appeared that Crabp1 was most highly expressed in the pituitary gland, then secondly in the hippocampus and finally followed by the adrenal gland. We then monitored the expression of several key genes in the hypothalamus and pituitary glands that are known to be involved in HPA axis regulation (Figure 3B–E). In the hypothalamus gland, CRH and vasopressin (AVP) act to stimulate ACTH production. Clearly, there was no difference in the expression of CRH (Figure 3B; T(9) = 0.895, *p* = 0.394) and AVP (Figure 3B; T(9) = 0.252, *p* = 0.806) between WT and CKO mice. In the pituitary gland, the gene POMC generates the precursor of ACTH, important for stimulating adrenal gland, whereas corticotropin-releasing hormone receptor 1 (CRHR1) receives the stimulating signal CRH from hypothalamus gland. The data showed that CKO mice had a significantly lowered expression level of POMC (Figure 3C; T(10) = 3.284, *p* < 0.01), but not CRHR1 (Figure 3C; T(10) = 0.278, *p* = 0.787). In CKO mice, the significant reduction in POMC level is consistent with the lowered corticosterone level. We then assessed the expression of GR (Gene name: NR3C1) and FKBP5, the principal proteins regulating feedback inhibition. Importantly, while there was no difference in the expression of GR in hypothalamus gland (Figure 3D; T(10) = 1.682, *p* = 0.123) or pituitary glands (Figure 3E; T(10) = 1.84, *p* = 0.096) between WT and CKO mice, FKBP5 mRNA level was significantly reduced in both hypothalamus gland (Figure 3D; T(9) = 2.386, *p* < 0.05) and pituitary glands (Figure 3E; T(10) = 3.339, *p* < 0.01) of CKO mice. Furthermore, at the protein level FKBP5 was also significantly reduced in the hypothalamus gland of CKO mice (Figure 3F,G; T(4) = 2.805, *p* < 0.05). Since FKBP5 regulates feedback inhibition by suppressing GR signaling, downregulation of FKBP5 in CKO mice would result in their enhanced sensitivity (hypersensitivity) of feedback inhibition in the HPA axis, and thus a lowered stress hormone level. These results are consistent with the somewhat dampened anxiety-like behavior and stress response in CKO mice (Figure 1 and Figure 2). The results prompted us to carefully examine whether FKBP5 could be a target of the action of Crabp1 in the pituitary or hypothalamus glands.

### 2.4. Altering Crabp1 Level Changed the Expression of FKBP5 in a Pituitary Gland Cell Line AtT20

To further evaluate the potential impact of Crabp1 on the expression of FKBP5 in either the hypothalamus or pituitary glands, we employed a well-established pituitary gland cell line AtT20 that expressed Crabp1 endogenously. We examined the effects of altering Crabp1 level on FKBP5, as well as key genes for HPA regulation in this in vitro model, by over-expression and silencing Crabp1 and then monitoring relevant gene expression in these cells (Figure 4A–H). As predicted, elevating Crabp1 level (by over-expression) increased the expression of FKBP5 and CHRH1, but not GR (Figure 4A, T(10) = 21.28, *p* < 0.01; Figure 4B, T(10) = 3.783, *p* < 0.01; Figure 4C, T(10) = 2.952, *p* < 0.05; Figure 4D, T(10) = 0.6888, *p* = 0.5066). On the contrary, lowering Crabp1 level (by silencing) reduced the expression of FKBP5 and CHRH1, but not GR (Figure 4E, T(10) = 3.86, *p* < 0.01; Figure 4F, T(10) = 3.569, *p* < 0.01; Figure 4G, T(10) = 3.514, *p* < 0.01; Figure 4H, T(10) = 0.3695, *p* = 0.7195). At the protein level, we further validated that the over-expression of Crabp1 significantly increased FKBP5 expression in the AtT20 cell line. (Figure 4I,J; T(4) = 5.362, *p* < 0.01). It was interesting that silencing Crabp1 for 48 h in this culture system caused only a small, but significant, down-regulation in FABP5 expression (Figure 4F, 41%, T(10) = 3.569, *p* = 0.0051). This could be due to that FKBP5 expression was regulated by multiple factors/signaling pathways; Crabp1 merely represented one regulatory pathway. Nevertheless, these data clearly demonstrated the molecular effects of Crabp1 in pituitary gland cells, that altering Crabp1 level impacted the expression of genes related to HPA axis regulation. In particular, the expression of one key regulator in feedback inhibition, FKBP5, was profoundly affected.

### 2.5. DEX and atRA Increased Crabp1 Expression in AtT20 Cell

We have previously reported that the mouse *Crabp1* gene could be regulated by atRA in fibroblast cells [20]. It was of interest to determine whether this gene could also be regulated by RA or a stress hormone more relevant to HPA regulation in pituitary gland cells. Thus, we assessed the Crabp1 protein level in AtT20 cells treated with control, 2 μM DEX or 100 nM CRH for 24 h. (Figure 5A) to mimic stress induction in the pituitary gland. Interestingly, Crabp1 level was elevated only by DEX treatment, suggesting that *Crabp1* expression was more sensitive to direct corticosterone regulation. The effect of atRA was determined using cultures supplemented with dextran-coated charcoal (DCC)-stripped FBS in the experiment (Figure 5B,C). After 24 h, treatment with 100 nM atRA, or 2 μM DEX, Crabp1 level indeed was significantly elevated.

These results confirmed that both a stress hormone, DEX, and atRA could elevate the Crabp1 level in pituitary gland cells, supporting a physiological relevance of Crabp1 to the stress response. Specifically, it could respond to stress by elevating its expression, and modulate the stress response by affecting the efficiency of HPA axis feedback response. The physiological relevance of RA induction of this gene in pituitary gland cells remains to be determined further (see Discussion).

## 3. Discussion

This is the first study uncovering the physiological relevance of Crabp1 to the stress response. Crabp1 responds to stress by elevating its expression and plays a functional role by modulating the HPA axis. One key regulatory protein in the HPA axis, FKBP5, appears to be one of the targets of Crabp1 in the pituitary and hypothalamus gland.

Consistently, Crabp1 is highly expressed in hypothalamus and pituitary glands in the central nervous system. Using the *Crabp1* gene knockout mouse model (CKO), we first observed significantly reduced acute stress-induced anxiety-like behaviors, accompanied by a much-lowered corticosterone level. The DEX/CRH test revealed their enhanced feedback inhibition in the HPA axis, suggesting that deleting Crabp1 further enhanced CKO animals’ feedback regulation in the HPA axis. As a result, their principal stress hormone, corticosterone, was suppressed under acute stress; therefore, CKO mice exhibited significantly reduced anxious behaviors. This is the first example where an RA signaling mediator, Crabp1, plays a functional role in modulating the endocrine system (specifically homeostasis of the HPA axis), as well as corresponding stress-induced behaviors/responses. Given the extremely high conservation of the *Crabp1* gene (almost 100% amino acid sequence conservation among various species including humans), it would be of interest to examine any association of *Crabp1* gene in human patients with endocrine disorders or stress-induced behavioral responses/changes (see later for more discussion).

Our targeted screening of the key gene expression profiles of the HPA axis in WT and CKO mice showed a dramatically and consistently altered gene, *Fkbp5* [10]. We found that the expression of FKBP5 was significantly decreased, when comparing CKO to WT mice, in their hypothalamus and pituitary glands where Crabp1 was most highly expressed. The effect of altering Crabp1 on FKBP5 expression was confirmed in a physiologically relevant in vitro culture model, a pituitary gland cell line AtT20 which expressed endogenous Crabp1. We confirmed that increasing or decreasing Crabp1 expression correspondingly altered FKBP5 level in this culture model.

FKBP5 is known to suppress GR signaling thereby regulating feedback inhibition in the HPA axis [9,10,21,22]. Tissue-specific knockout of FKBP5 in Sim1^+^ cells in the paraventricular nucleus dampened the acute stress response and increased GR signaling activity. In contrast, FKBP5 overexpression in the paraventricular nucleus led to chronic hyperactivity of the HPA axis [23]. Reduction in FKBP5 level, as a result of deleting Crabp1, would result in increased GR signaling activity, and thus result in enhanced feedback inhibition and a correspondingly lowered corticosterone level. This is consistent with changes observed in these CKO animals’ stress behaviors, as assessed by specific stress-induced behavioral patterns. It is tempting to speculate that Crabp1 regulates animals’ stress response and HPA axis homeostasis by, at least partially, maintaining (or up-regulating) FKBP5 expression to suppress GR signaling which regulates critical steps of HPA feedback inhibition.

Our data also showed that a glucocorticoid analog, DEX, or atRA, could increase Crabp1 expression in the pituitary gland cell line. To this end, it has been shown that acute high-dose DEX treatment in a dental procedure significantly decreased CpG island methylation on the *Crabp1* gene promoter [15], suggesting that GR or acute stress could upregulate Crabp1 expression. Of noteworthy mention is that patients with Cushing’s syndrome (hypercortisolism) generally showed reduced levels of DNA methylation in genes related to RAR binding, RAR activity, and RXR activity [24]. Glucocorticoid replacement therapy also resulted in hypomethylation in the *Fkbp5* gene [24]. Moreover, studies in mice have showed that long-tern exposure to Glucocorticoid reduced DNA methylation of *Fkbp5* in the hippocampus and hypothalamus, and the demethylation was associated with anxiety-like behavior [25,26]. This evidence highlighted a correlation between increased CRABP1 and FKBP5 level, and enhanced RA signaling activity in glucocorticoid exposure. Comprehending this relationship is helpful to the understanding of the mechanism of hypervitaminosis A-induced psychiatric disorders [1,2,3].

DEX increased Crabp1 level, which could disturb feedback inhibition of the HPA axis, allowing stress response hormones to be efficiently elevated in the face of stress for healthy animals. The CKO mouse model might mimic a diseased condition where the feedback regulation of stress response cannot be efficiently regulated, as reflected on their less anxious phenotype even in the face of stress. Interestingly, chronic unpredictable mild stress decreased *Crabp1* expression in the hypothalamus gland of rats [27]. This would suggest that the Crabp1 level might be lower in patients suffered from chronic stress (or depression). Thus, the regulation of *Crabp1* gene expression in HPA axis could be much more complex and could vary between acute versus chronic stress conditions. It would be interesting to examine whether *Crabp1* gene appears in the list of genes with disease association, such as for those patients suffering from anxiety or depression disorder.

As for the effect of RA, it has been shown that atRA could regulate Crabp1 expression in fibroblast culture and adipocytes [17,19,20,28,29,30]. In this current study, our results showed that atRA could also up-regulate *Crabp1* in pituitary gland cells. Therefore, it would also be interesting to examine whether there is any relationship of endogenous RA biosynthesis/degradation pathways with human patients of endocrine or stress-related disorders. To this end, clinical and pre-clinical data showed that RA treatment (atRA and 13-cis RA) could be related to anxiety and depression disorders, and it has been suggested that these compounds might impair the HPA axis [1,2,3]. In clinical settings, 13-cis RA is typically used in oral treatment for severe acne, and atRA is mainly used to manage acute promyelocytic leukemia [31]; their effects are typically attributable to the actions of RAR/RXR13]. This current study reveals that Crabp1 may also contribute to RA-induced side effects, such as in altering the stress response and HPA axis regulation.

As for the molecular mechanism underlying the effect of Crabp1 on FKBP5, it remains to be studied. Our previous studies have identified at least two cytosolic signaling pathways that could be targeted by Crabp1, the ERK (MAPK) and CaMKII pathways [14,18,32,33]. The effect of Crabp1 on ERK pathway is elicited mainly in proliferating cells such as tumor or stem cells [17,19,32,34], whereas that on CaMKII pathway is elicited mainly in differentiated cells such as cardiomyocyte and neurons [13,14,33]. Further studies are needed to determine whether either one of these two pathways is involved in the regulation of *Fkbp5* gene expression in pituitary gland cells; alternatively, Crabp1 may target entirely different signaling pathways in these specialized cells to regulate FKBP5 level, which remains to be studied.

## 4. Materials and Methods

### 4.1. Mice

Male WT and Crabp1 knockout (CKO) mice were used for all experiments. CKO mice were generated as described in Lin et al. [19]. Three-month-old WT (C57BL/6) and CKO mice were used in these studies. The mice were housed and bred in the University of Minnesota Research Animal Resources facilities. Animals were housed in temperature-controlled (22 ± 1 °C) housing on a 14/10 light/dark cycle (lights on/off at 0600/2000) with ad libitum food and water. The experimental procedures were conducted according to National Institutes of Health guidelines and were approved by the University of Minnesota Institutional Animal Care and Use Committee.

### 4.2. Behavioral Assessments

All the mice were handled for 3 days in same room as the behavioral experiments were conducted in order to reduce background stress and anxiety. The experiments were performed between 11 am and 5 pm. The apparatuses were cleaned with 70% ethanol and air-dried between each trial.

#### 4.2.1. Open Field Test

Open field testing was performed as described in [19] with slight modifications. Briefly, mice were placed in an open field comprised of a Plexiglass box with 40 × 40 × 25 cm dimensions in a dimly lit environment. The center area was defined as a 20 × 20 cm square. Each mouse was allowed 5 min of free movement. Data were collected via a ceiling-mounted camera and analyzed using Any-maze^TM^ software (Stoelting Co., Wood Dale, IL, USA).

#### 4.2.2. Elevated plus Maze Test

The Elevated plus Maze is a widely used animal model of anxiety that is based on conflicting tendencies [35]. The apparatus (Med-Associated, St. Albans, VT, USA) consists of two open and two enclosed arms elevated from the floor. Mice were placed into the center of the apparatus at the beginning of the session; the number of arm entries and the amount of time spent in the open and closed arms are recorded for 5 min. The data were collected via a ceiling-mounted camera and analyzed using EthoVision XT (Noldus, Leesburg, VA, USA).

### 4.3. Corticosterone Level

Blood samples were collected from the submandibular vein for no more than one minute into a microtube without anticoagulant and kept on ice for 30 min. Serum was collected by centrifugation at 1000× *g* for 15 min and stored at −80 °C until analysis. Corticosterone concentration was determined by ELISA (Caymanchem, MI, USA) according to the manufacturer’s protocol.

#### 4.3.1. Acute Restraint Test

For the basal condition, blood samples were collected from WT and CKO mice at 2 pm. For the acute restraint test, mice were restrained using the TailVeiner Restrainer TV-150 device (Braintree Scientific, MA, USA) for 15 min. Blood samples were then immediately collected.

#### 4.3.2. Combined DEX/CRH Test

The combined DEX/CRH test was performed as described in with slight modifications [36]. The mice were intraperitoneally (IP) injected with a low dose DEX (0.05 mk/kg) at 9 am. Six hours after DEX injection, the blood samples were collected. Then, mice received an injection of CRH (0.2 mg/kg; IP). A total of 30 min after CRH injection, blood samples were collected.

### 4.4. Tissue Sample Collection

Mice were euthanized by CO_2_. Immediately, the pituitary gland, hypothalamus gland, hippocampus, and adrenal gland were collected and processed for downstream RT-qPCR and Western blot experiments.

### 4.5. AtT20 Cell Culture, Crabp1 Overexpression and Crabp1 Silencing

AtT20, cells were maintained with Dulbecco’s modified eagle’s medium (DMEM; Gibco) supplemented with 10% fetal bovine serum (Gibco/Thermofisher, Waltham, MA, USA), 1% penicillin and streptomycin at 37 °C in 5% CO2 incubator. One day before experiments, 6.0 × 10^5^ AtT20 cells per well of a 6-well plate were seeded in 2 mL of supplemented DMEM medium. Mouse Crabp1 cDNA was cloned into pCMX-PL1 plasmid with Flag-tag. Crabp1 over-expression in AtT20 was achieved by transfecting Flag-Crabp1 using Lipofectamine 3000 (Invitrogen/Thermofisher, Waltham, MA, USA) according to manufacturer’s instructions. Endogenous Crabp1 expression was silenced using the HiPerfect transfection reagent (Qiagen, Germantown, MD, USA) as described in [29]. Crabp1 siRNAs and negative control siRNA were purchased from Qiagen. The sequence of mouse Crabp1 siRNA are 5′-CACGTGGGAGAATGAGAACAA-3′ and 5′-CAGCTTGTTCCTGCTTCATGA-3′. Cells were harvested 48 h later for downstream RT-qPCR and Western blot experiments.

### 4.6. Western Blotting

Western blotting was performed as described [18]. Hippocampus, hypothalamus gland, pituitary gland, and adrenal gland were sonicated in RIPA buffer with proteinase inhibitors (Thermofisher, Waltham, MA, USA). An amount of 50 ug of protein was loaded for SDS-PAGE. Primary antibodies, Crabp1 (1:1000; Sigma-Aldrich, St. Louis, MO, USA), FKBP5 (1:1000; Cell Signaling Technology, Danvers, MA, USA), and β-actin (1:2500; Santa Cruz Biotechnology, Dallas, TX, USA).

### 4.7. RT-qPCR

RNA from isolated brain tissues was extracted by TRIzol Reagent (ThermoFisher). RNA concentration was measured with NanoDrop and cDNA was synthesized by High-Capacity cDNA Reverse Transcription Kit (ThermoFisher). Quantitative RT-PCR (qPCR) was performed using the SYBR™ Green PCR Master Mix (ThermoFisher) and following primers which are listed in Table 1. GAPDH was used for normalization. qPCR was conducted on Mx3000P QPCR Systems (Agilent, San Diego, CA, USA).

### 4.8. Statistical Analysis

Data were analyzed using Student’s t test (WT V.S. CKO; Vector V.S. Crabp1; siCtrl V.S. siCrabp1). Statistical analyses were performed using Prism 8.0 (GrapgPad, CA). For three group comparisons, a one-way analysis of variance (ANOVA) followed by Dunnett’s post hoc test was performed. All tests were performed at a significance level of *p* ≤ 0.05, and data are presented as mean ± standard error or mean ± SD of the mean as indicated.

## Figures and Tables

**Figure 1 ijms-22-12240-f001:**
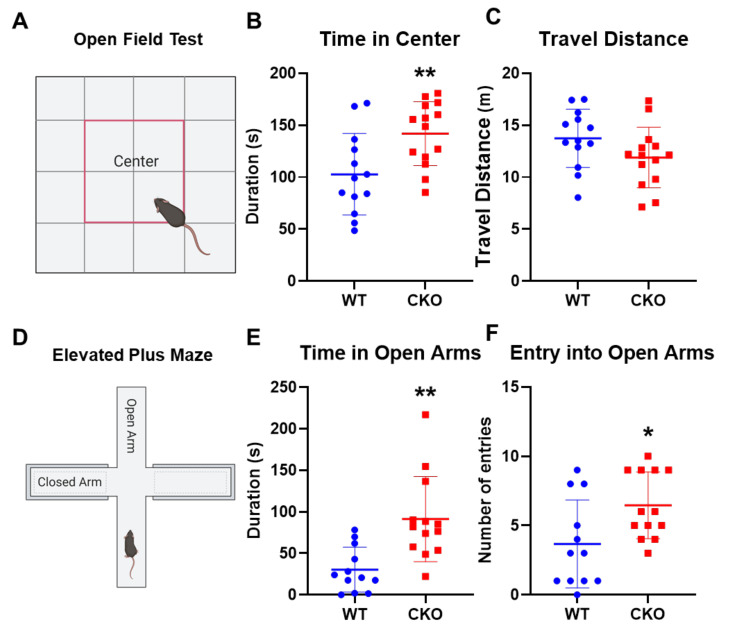
CKO mice exhibit reduced anxiety-like behaviors. (**A**–**C**) Open field test. The duration in the central area (**B**) and the total travel distance (**C**) in WT (*n* = 13) and CKO (*n* = 14) mice. (**D**,**E**) Elevated plus maze test. The duration in open arms € and numbers of entry into open arms (**F**) of WT (*n* = 12) and CKO (*n* = 13) mice. Results are presented as means  ±  SD, * *p*  ≤  0.05, ** *p*  ≤  0.01, compared with the WT group.

**Figure 2 ijms-22-12240-f002:**
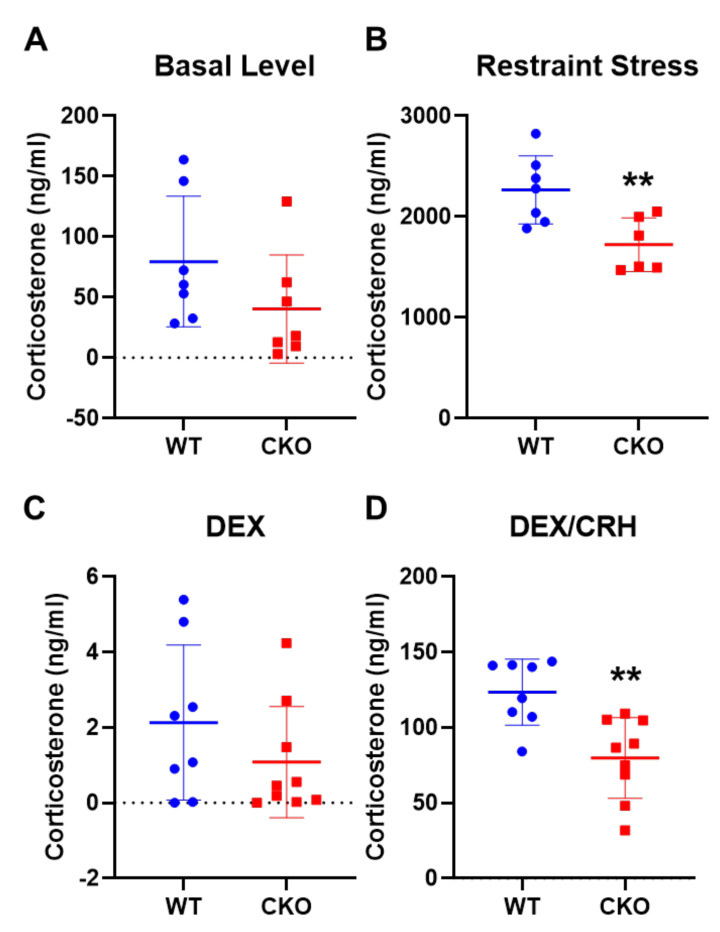
CKO mice have reduced corticosterone levels after restraint stress and in DEX/CRH testing. (**A**,**B**) Corticosterone concentration in basal (**A**) and restraint stressed conditions (**B**) in WT (*n* = 7) and CKO (*n* = 7) mice. (**C**,**D**) DEX/CRH test. The corticosterone level after DEX treatment (**C**) and then CRH challenge (**D**) in WT (*n* = 13) and CKO (*n* = 14) mice. Results are presented as means  ±  SD, ** *p*  ≤  0.01, compared with the WT group.

**Figure 3 ijms-22-12240-f003:**
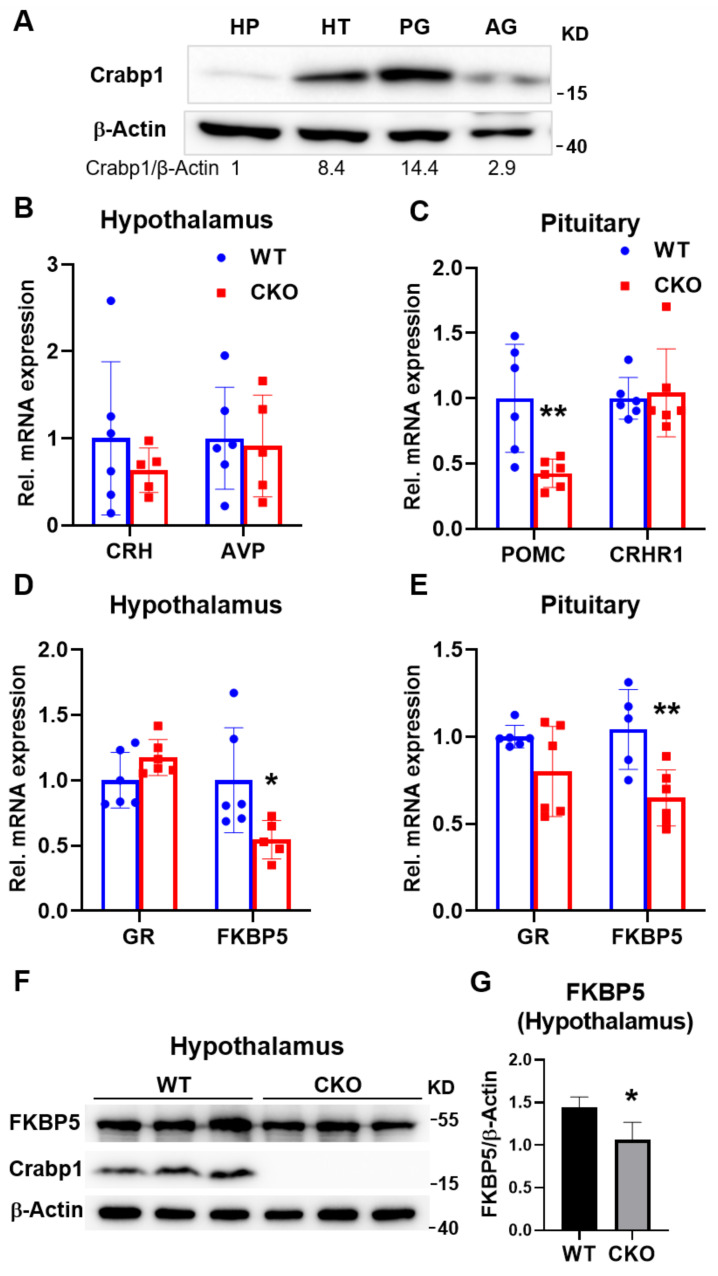
Changes in the expression patterns of key genes relevant to the regulation of HPA axis in hypothalamus and pituitary gland tissues. (**A**) Crabp1 protein levels in the hippocampus (HP), hypothalamus gland (HT), pituitary gland (PG), and adrenal gland (AG). (**C**,**D**) Changes in gene expression in the hypothalamus gland (**B**,**D**) and pituitary gland (**C**,**E**) of WT (*n* = 6) and CKO (*n* = 6) mice. (**F**,**G**) FKBP5 protein levels in the hypothalamus gland of WT (*n* = 3) and CKO (*n* = 3) mice. β-actin was used as a loading control. Western blot quantification shown in (**G**). Results are presented as means  ±  SD, * *p*  ≤  0.05, ** *p*  ≤  0.01, compared with WT group.

**Figure 4 ijms-22-12240-f004:**
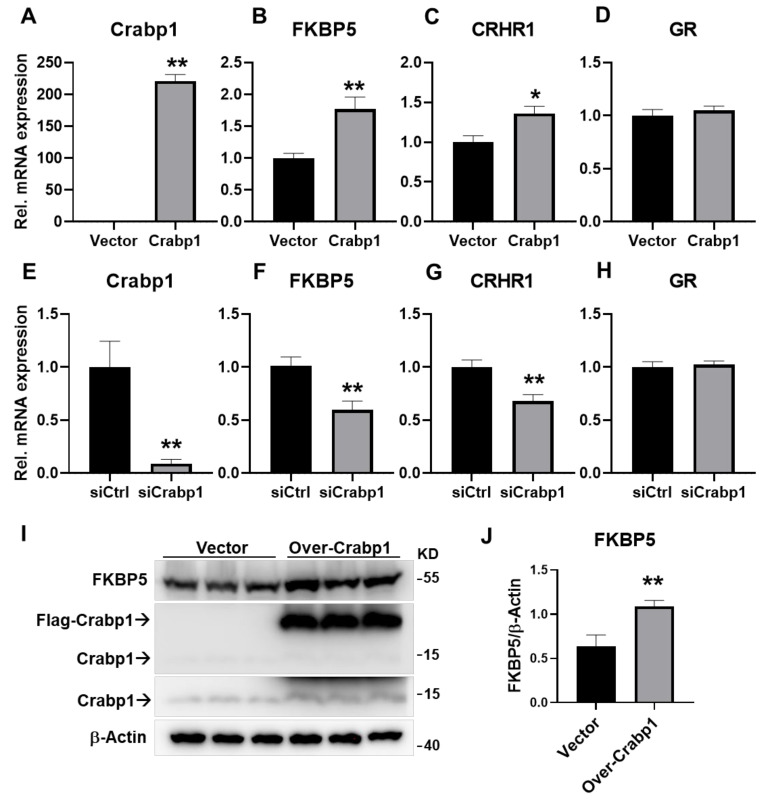
Changes in mRNA levels of Crabp1, FKBP5, CRHR1, and GR and FKBP5 protein levels in AtT20 cells with elevated (over-expression) or lowered (silencing) Crabp1 levels. (**A**–**D**) mRNA levels of key genes in Crabp1-elevated AtT20 cell (*n* = 6/groups). (**E**–**H**) mRNA levels of key genes in Crabp1-lowered AtT20 cell (*n* = 6/groups). (**I**,**J**) Changes in FKBP5 protein levels upon Crabp1-expression. Labels are indicated as “Vector” = Empty vector control; “Over-Crabp1” = Crabp1 overexpression; “# Crabp1” = endogenous Crabp1 imaged with a short (top) or long exposure time (bottom). β-actin was used as a loading control. Results are presented as means  ±  SEM, * *p*  ≤  0.05, ** *p*  ≤  0.01, compared with the Ctrl group.

**Figure 5 ijms-22-12240-f005:**
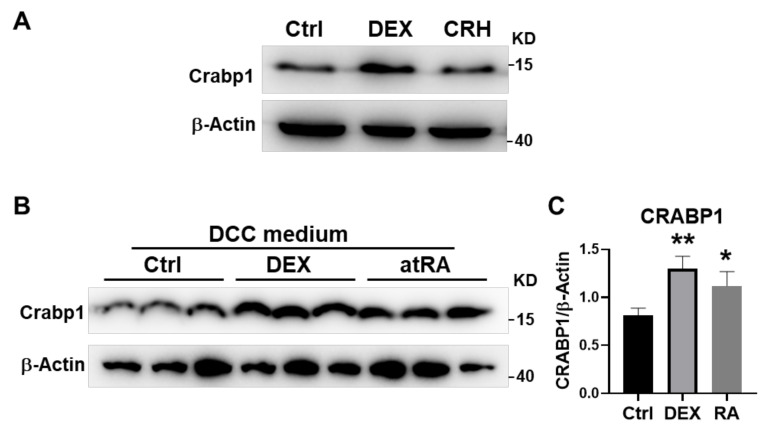
Crabp1 protein levels in AtT20 cells. (**A**) Crabp1 level in AtT20 cells after 24 h DEX or CRH treatment. (**B**) Crabp1 level in AtT20 cells grown in DCC FBS-supplemented medium after 24 h. RA or DEX treatment. Western blot quantification shown in (**C**). β-actin was used as a loading control. Results are presented as means  ±  SEM, * *p*  ≤  0.05, ** *p*  ≤  0.01, compared with the Ctrl group.

**Table 1 ijms-22-12240-t001:** Primer List.

Gene	Forward Primer	Reverse Primer
*CRH*	GGAATCTCAACAGAAGTCCCGC	CTGCAGCAACACGCGGAAAAAG
*AVP*	GCTACTTCCAGAACTGCCCAAG	CAGCAGATGCTTGGTCCGAAGC
*POMC*	CCATAGATGTGTGGAGCTGGTG	CATCTCCGTTGCCAGGAAACAC
*CRHR1*	CGCAAGTGGATGTTCGTCTGCA	TCCAGGACGTTTGCCAAACCAG
*GR (Nr3c1)*	TGGAGAGGACAACCTGACTTCC	ACGGAGGAGAACTCACATCTGG
*FKBP5*	GATTGCCGAGATGTGGTGTTCG	GGCTTCTCCAAAACCATAGCGTG
*Crabp1*	CGGAGATCAACTTCAAGGTCGG	CCCTCAAGAAGTGTCTGTGTGC
*GAPDH*	CATCACTGCCACCCAGAAGACTG	ATGCCAGTGAGCTTCCCGTTCAG

## Data Availability

All the data of this study are available from the corresponding author upon reasonable request.

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
