# Peer review of "Crabp1 Modulates HPA Axis Homeostasis and Anxiety-like Behaviors by Altering FKBP5 Expression"

_ijms, 2021, doi:10.3390/ijms222212240_

Round 1
Reviewer 1 Report
The authors addressed a very relevant issue, that is the characterization of mechanisms underlying the modulatory action of retinoic acid in the activation of hypothalamus- pituitary-adrenal (HPA) axis. The study shows, for the first time, that Crabp1 regulates feedback inhibition of the HPA axis by modulating the expression of FKBP5. Further, retinoic acid and stress can increase Crabp1 level, which would up-regulate FKBP5 thereby de-sensitizing feedback inhibition of HPA axis (via decreasing glucocorticoid receptor signaling) and increasing the risk of stress-related disorders. The results are very interesting and potentially relevant in a translational perspective. However, some criticisms remain to be addressed.
Major points
To confirm the changes in the expression of FKBP5 in CKO mice compared with WT ones, as well as in AtT20 cells overexpressing Crabp1 in comparison with ones silenced for Crabp1, Western blotting experiments had to be carried out. The conclusions would be strenghtened by these additional experiments.
The authors should provide more details about cell culturing experiments.
How the authors explain that in AtT20 cells Crabp1 silencing is not able to suppress FABP5 expression or inhibit ita t a greater extent than that observed in their experiments?In this regard, the authors should describe in detail the Crabp1 silencing protocol used for their experiments, as well as the transfectiuon protocol used for Crabp1 overexpression experiments.
About alterations of DNA methylation, including the promoter region of FKBP5 and retinoic acid receptor, induced by hypercortisolism, it would be worthy mention the interesting paper by Glad and co-workers (Sci Rep 2017).
Minor points
In Table I please correct GAPDG by replacing with GAPDH.
The words pituitary and adrenal should be replaced by pituitary gland and adrenal glan when not used in the set hypothalamic-pituitary-adrenal axis.
In the Introduction mention in full atRA (all-trans Retinoic acid?) at the first use.
Several typos throughout the text have to be corrected.
Reference 32 is the double of ref. 17.
Author Response
Reviewer 1
The authors addressed a very relevant issue, that is the characterization of mechanisms underlying the modulatory action of retinoic acid in the activation of hypothalamus- pituitary-adrenal (HPA) axis. The study shows, for the first time, that Crabp1 regulates feedback inhibition of the HPA axis by modulating the expression of FKBP5. Further, retinoic acid and stress can increase Crabp1 level, which would up-regulate FKBP5 thereby de-sensitizing feedback inhibition of HPA axis (via decreasing glucocorticoid receptor signaling) and increasing the risk of stress-related disorders. The results are very interesting and potentially relevant in a translational perspective. However, some criticisms remain to be addressed.
Major points
To confirm the changes in the expression of FKBP5 in CKO mice compared with WT ones, as well as in AtT20 cells overexpressing Crabp1 in comparison with ones silenced for Crabp1, Western blotting experiments had to be carried out. The conclusions would be strenghtened by these additional experiments.
Response: We have added relevant Western blot data, listed below:
1). The expression of FKBP5, Crabp1 and b-Actin in hypothalamus were added in Fig 3F. Pituitary gland is too small for performing Western blotting (3 pituitary gland for 1 WB); However, we showed data for FKBP5 mRNA expression in pituitary in Fig 3E.
2.) The expression of FKBP5, Crabp1 and b-Actin in AtT20 cells with over-expressed Crabp1 were added in Fig 4I.
3.) In AtT20 cells, reduction of FKBP5 protein level under Crabp1 silencing (for 2 days) was not significant. This could be due to the fact that change in protein levels using siRNA could take a longer period of time, as compared to the effect on mRNA expression. This is further compounded by the time needed to first silence Crabp1 (for effective reduction in both mRNA and protein levels). Therefore, in this experiment (silencing for 48 hrs), we showed only the effect on mRNA level (Fig. 4F).
The authors should provide more details about cell culturing experiments.
Response: In the Materials and Methods Section 4.5 we have added more details regarding AtT20 cell culture, Crabp1 overexpression and Crabp1 silencing experiments, including information for plasmids, siRNAs, transfection methods, and relevant references.
How the authors explain that in AtT20 cells Crabp1 silencing is not able to suppress FABP5 expression or inhibit ita t a greater extent than that observed in their experiments? In this regard, the authors should describe in detail the Crabp1 silencing protocol used for their experiments, as well as the transfectiuon protocol used for Crabp1 overexpression experiments.
Response:
1). In Fig 4F, Crabp1 silencing did suppress FABP5 mRNA expression, although not as robust as one would predict if Crabp1 provides the only one regulatory pathway. This is partially explained in above response, point 2. Additionally, FKBP5 expression can be regulated by multiple factors/pathways [1]. Therefore, while Crabp1 clearly contributes to the regulation of FKBP5 mRNA expression, it remains unclear to what extent Crabp1 contributes to overall regulation of FKBP5, and the mechanism remains to be further examined. This current study focuses on the first report of such a novel role of Crabp1 in FKBP5 regulation; more mechanistic studies clearly are needed to dissect all the mechanisms pertinent to FKBP5 regulation, as well as the question, to what extent Crabp1 could contribute to overall regulation of FKBP5. This is discussed in the text relevant to Fig. 4.
2) We have added more details about transfection protocols, Crabp1 silencing and overexpression in the Material and Methods section. Please see Materials and Methods Section 4.5.
- Zannas, A.S.; Wiechmann, T.; Gassen, N.C.; Binder, E.B. Gene-Stress-Epigenetic Regulation of FKBP5: Clinical and Translational Implications. Neuropsychopharmacology 2016, 41, 261–274.
About alterations of DNA methylation, including the promoter region of FKBP5 and retinoic acid receptor, induced by hypercortisolism, it would be worthy mention the interesting paper by Glad and co-workers (Sci Rep 2017).
Response: Thank you for the important reference. We have added the reference into discussion.
Minor points
In Table I please correct GAPDG by replacing with GAPDH.
Response: We have corrected this typo.
The words pituitary and adrenal should be replaced by pituitary gland and adrenal gland when not used in the set hypothalamic-pituitary-adrenal axis.
Response: We have corrected the writing.
In the Introduction mention in full atRA (all-trans Retinoic acid?) at the first use.
Response: All-trans retinoic acid has been mentioned in full and corrected in the introduction.
Several typos throughout the text have to be corrected.
Response: The manuscript has been proofread carefully.
Reference 32 is the double of ref. 17.
Response: We have removed the duplicate reference.
Reviewer 2 Report
In their manuscript, Lin and colleagues report that Crabp1 regulates feedback inhibition of the HPA axis via modulating FKBP5 expression. Moreover, they speculate that RA and stress can increase Crabp1 level, which would up-regulate FKBP5 thereby de-sensitizing feedback inhibition of HPA axis and increasing the risk of stress-related disorders. The manuscript is well-written. The experiments are well designed. The results are interesting and would be of general interest to the field of stress research. However, some minor points should be addressed before publication:
- Results (2.1): Authors report: “Crabp1 knockout (CKO) mice were less anxious. This statement need to be corrected because “less anxious” is a terminology used for humans and not for rodents. It is more corrected Crabp1 knockout (CKO) mice showed a descrease anxiety like behavior.
- Figure 1 results: Regarding the open field data, Authors report only the time spent in the center and the travel distance. It would be nice if authors would also show the number of entries in the center of the OF.
- Results (2.3 and 2.5; figures 3 and 5): Authors only report western blotting images without showing data as bar graphs. Moreover, authors did not perform statistical analysis for these data. Obviously, Authors need to sort out this critical point.
- Discussion: Authors brilliantly discover for the first time that Crabp1 regulates feedback inhibition of the HPA axis via modulating FKBP5 expression. However, Authors also found that deletion of Crabp1 leads to a downregulation of FKBP5 in the hypothalamus and to a decreased anxiety-like behavior. These results should be better discussed in relation to the available literature in this field. In particular, in line with these data, the downregulation of FKBP5 in the hypothalamus has been recently found and linked to traumatic stress resilience (PMID: 33392367). Moreover, it has recently been found that Fkbp5 shapes the acute stress response in the paraventricular nucleus of the hypothalamus (PMID: 33649453). Authors might want to discuss these recent papers in order to improve the interpretation of their results.
- Materials and Methods: Authors need to report if they used males, females or both for this study. Generally, there is a lack of references.
Author Response
Reviewer 2. Comments and Suggestions for Authors
In their manuscript, Lin and colleagues report that Crabp1 regulates feedback inhibition of the HPA axis via modulating FKBP5 expression. Moreover, they speculate that RA and stress can increase Crabp1 level, which would up-regulate FKBP5 thereby de-sensitizing feedback inhibition of HPA axis and increasing the risk of stress-related disorders. The manuscript is well-written. The experiments are well designed. The results are interesting and would be of general interest to the field of stress research. However, some minor points should be addressed before publication:
- Results (2.1): Authors report: “Crabp1 knockout (CKO) mice were less anxious. This statement need to be corrected because “less anxious” is a terminology used for humans and not for rodents. It is more corrected Crabp1 knockout (CKO) mice showed a descrease anxiety like behavior.
Response: We have corrected this statement to “Crabp1 knockout (CKO) mice show decreased anxiety-like behavior” (see Results 2.1). - Figure 1 results: Regarding the open field data, Authors report only the time spent in the center and the travel distance. It would be nice if authors would also show the number of entries in the center of the OF.
Response: Thank you for the suggestion. In our study, there was no significant difference between WT and CKO mice in the number of entries into the center during the OFT (Fig A). Therefore, we proposed that Crabp1 KO did not affect the exploratory behavior and activity of the mice; as such, these data were not included in the manuscript. Please see the right figure below (Fig B). The heat map and position tract data did suggest that WT mice prefer exploring the corner areas as compared to the CKO mice (Fig B). We reasoned that time spent in the center provided a more meaningful index to assess anxiety-like behavior in OFT as compared to the number of entries in the center (Fig B right; position tract). Moreover, the elevated plus maze data also supported this result of Time in Center of OFT.
(A) (B)
- Results (2.3 and 2.5; figures 3 and 5): Authors only report western blotting images without showing data as bar graphs. Moreover, authors did not perform statistical analysis for these data. Obviously, Authors need to sort out this critical point.
Response:
1). Thank you for the points.
Fig 3A showed a representative image for the relative Crabp1 expression levels in tissues of the HPA axis, therefore no quantification was shown.
2). We have added the statistical analysis in the Results Section 2.4.
- Discussion: Authors brilliantly discover for the first time that Crabp1 regulates feedback inhibition of the HPA axis via modulating FKBP5 expression. However, Authors also found that deletion of Crabp1 leads to a downregulation of FKBP5 in the hypothalamus and to a decreased anxiety-like behavior. These results should be better discussed in relation to the available literature in this field. In particular in line with these data, the downregulation of FKBP5 in the hypothalamus has been recently found and linked to traumatic stress resilience (PMID: 33392367). Moreover, it has recently been found that Fkbp5 shapes the acute stress response in the paraventricular nucleus of the hypothalamus (PMID: 33649453). Authors might want to discuss these recent papers in order to improve the interpretation of their results.
Response: Thank you for the supportive reference. We have added reference (PMID:33649453) into discussion. The interesting reference (PMID: 33392367) is related to the long-term stress adaptation and FKBP5 expression. However, this current study focuses on Crabp1 regulating FKBP5 in acute stress response, therefore we felt this interesting reference may not be quite appropriate for our current study.
- Materials and Methods: Authors need to report if they used males, females or both for this study. Generally, there is a lack of references.
Response:
1). Male mice were used for all experiments. We described this in Materials and Methods Section 4.1.
2). We provided more references and details in the Materials and Methods.
Round 2
Reviewer 1 Report
The authors satisfactorily addressed all criticisms raised during the peer-review process and manuscript quality was improved. The revised paper is worthy of publication in IJMS.